# GNNUPDATER: ADAPTIVE SELF-TRIGGERED TRAINING FRAMEWORK ON DYNAMIC GRAPHS

## ABSTRACT

Adapting Graph Neural Networks (GNNs) to evolving, dynamic graph data presents a significant operational challenge. A critical yet understudied question is determining *when* to update these models to balance model freshness against computational training costs. This problem is particularly difficult in graph settings due to two key issues: *label delay*, where ground truth arrives long after predictions are made, and *hidden drift*, where structural dependencies propagate changes through multiple hops, causing unexpected performance degradation. We propose GNNUpdater, an adaptive framework that decides when to trigger GNN training. It overcomes the aforementioned challenges through two innovations: (1) a performance predictor that estimates model quality by measuring shifts in node embeddings, eliminating dependence on immediate ground-truth labels, and (2) a graph-aware update trigger that uses label propagation to detect widespread performance degradation across the graph. We implement GNNUpdater as a high-performance distributed streaming-GNN library for billion-edge dynamic graphs. Extensive experiments demonstrate that GNNUpdater either exceeds the performance of periodic, performance-based, and drift-detection baselines at comparable training cost or matches their performance with significantly reduced computational effort. The implementation can be found in the anonymous link: `https://anonymous.4open.science/r/GNNUpdater-B47D/`.

## 1 INTRODUCTION

In machine learning systems deployed in dynamic environments, models inevitably grow stale. Shifting data distributions, or *concept drift* Gama et al. (2014a), can degrade model performance, often silently. A critical operational question is therefore: *when should we update the model?* Updating too frequently is costly and resource-intensive, while updating too slowly leads to prolonged periods of poor performance. This necessitates intelligent triggers that balance model freshness with training costs. Common strategies include periodic retraining Azure (2025), updates triggered by performance drops Nigenda et al. (2022), and data distribution shifts Lu et al. (2018).

This challenge is especially acute for Graph Neural Networks (GNNs) operating on dynamic graphs. Real-world graphs, such as social networks, financial transaction logs, and e-commerce platforms, evolve continuously Wang et al. (2021a); Lin et al. (2022); Fan et al. (2019). While the problem of update timing is recognized in traditional ML NannyML (2023); Evidently (2023); Van Looveren et al. (2019), it remains significantly understudied for GNNs, despite the inherently dynamic nature of their data. The core question of **when** to update a GNN in a streaming graph environment is largely unexplored, yet has critical impacts for system performance and operational cost.

**Challenges.** Two fundamental challenges make effective update timing for GNNs particularly difficult (Fig. 1). First, **label delay**: Performance-based triggers require immediate ground truth, but labels in graph applications are often delayed. For instance, fraud labels require lengthy investigations Wang et al. (2021a), and recommendation feedback can take hours or days Ni et al. (2019). This gap allows performance degradation to go undetected. Second, **hidden drift**: Unlike independent data points in traditional ML, nodes in a graph are interconnected. A single change (e.g., a new transaction) can affect its immediate neighbors and, through the GNN's multi-hop aggregation mechanism Hamilton et al. (2017), alter the embeddings of nodes several hops away. This structural propagation can degrade the embeddings of even long-inactive nodes. Triggers that only monitor recent data will fail to detect this underlying drift until errors have already surfaced.

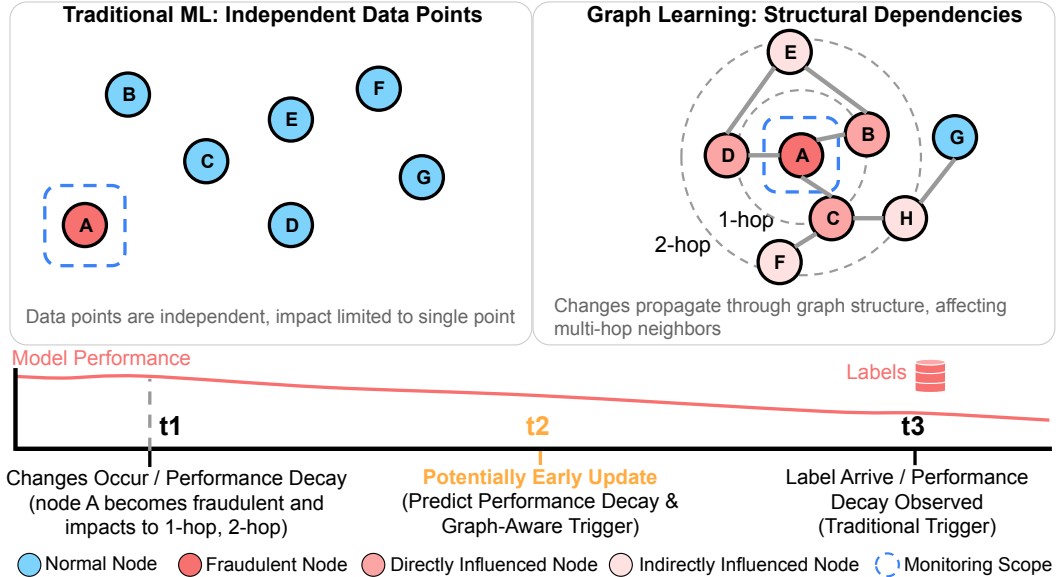

Figure 1: Challenges of GNN update timing. *Top*: In traditional ML (left), data points are independent. In graph learning (right), a change to one node propagates through its multi-hop neighborhood, causing *hidden drift*. *Bottom*: Performance decays at $t_1$, but labels only arrive at $t_3$ (label delay). A traditional trigger updates too late. Our graph-aware trigger monitors embedding shifts to predict performance decay, enabling an early update at $t_2$.

**Our Approach.** We propose GNNUpdater, an adaptive framework for triggering GNN updates on dynamic graphs. GNNUpdater is designed to overcome both label delay and hidden drift. Its core is a **performance predictor** (§3.1) that estimates task performance without ground-truth labels. It does so by measuring the shift in node embeddings relative to a stable reference period. The rationale is that significant changes to a GNN's output embeddings signal that the downstream model is encountering data it was not trained on, which typically degrades performance. We confirm a strong correlation between embedding shifts and task performance (e.g., a -0.96 Pearson correlation), validating this approach for timely degradation detection.

Building on this predictor, we introduce a **graph-aware update trigger** that monitors model quality across the entire graph. Each node is classified as either *normal* or *problematic*. As new data arrives, we predict the performance for active nodes and label them as problematic if the predicted performance falls below a threshold. This creates a semi-supervised learning problem on the graph, where a few nodes have fresh labels (normal/problematic) and most are unlabeled. We use label propagation Zhu & Ghahramani (2002) to diffuse these status labels across the graph structure. An update is triggered only when the proportion of problematic nodes crosses a global threshold, indicating widespread, systemic degradation. This graph-aware approach allows GNNUpdater to detect and react to hidden drift before it leads to catastrophic performance drops.

**Contributions.** We propose GNNUpdater, a novel framework that decides when to update GNNs on evolving graphs. It combines a label-delay tolerant performance predictor with a graph-aware trigger to balance accuracy and computational cost. We implement GNNUpdater in a high-performance, distributed streaming-GNN library designed for billion-edge dynamic graphs, which includes a custom dynamic graph storage system that significantly reduces operational overhead compared to standard frameworks like DGL Wang et al. (2019). Through extensive experiments on large-scale benchmarks, we show that GNNUpdater provides state-of-the-art results, outperforming periodic, performance-based, and traditional drift-detection baselines by either achieving higher accuracy for a similar cost or matching accuracy for a fraction of the training effort.

## 2 PRELIMINARIES AND PROBLEM FORMULATION

**Streaming Graphs.** We observe a sequence of temporal edge-update batches $\{\Delta G_t\}_{t=1}^{T}$, where $\Delta G_t = \{(u_i, v_i, t_i)\}_{i=1}^{n_t}$ consists of $n_t$ edges arriving in window $t$. Each node $v$ has a feature vector. The cumulative graph is $G_t = (V_t, E_t)$ with $V_t = \bigcup_{i \leq t} V(\Delta G_i)$, $E_t = \bigcup_{i \leq t} E(\Delta G_i)$. A subset $\mathcal{T}_t \subseteq V_t$ are *target nodes*: each $v \in \mathcal{T}_t$ has a true label $y_v$, which is generated in window $t$ but arrives after a $\tau$-step delay. We denote by $\mathcal{Y}_t = \{y_v | v \in \mathcal{T}_t\}$ the set of labels associated with batch $t$.

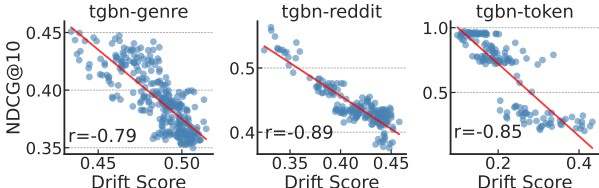

Figure 2: The embedding drift score shows a strong negative correlation with model performance across datasets for TGAT Xu et al. (2020a), with a Pearson correlation ($r$) of up to -0.89. The red line represents the linear regression fit.

Figure 3: The predicted performance closely matches the true performance, even when performance suddenly increases (caused by model updates).

**Concept Drift in Dynamic Graphs.** The evolving nature of streaming graphs gives rise to several types of concept drift, which can degrade model performance Lu et al. (2018). We identify two primary categories: (1) *Structural and Feature Drift* (a form of $p(x)$ drift), which includes changes to the graph topology or node features, and (2) *Semantic Drift* (a form of $p(y|x)$ drift), where the underlying relationship between graph features and target labels changes.

**GNN Inference and Training on Streaming Graphs.** We maintain a GNN with parameters $\theta$. At each time $t$ we (i) infer predictions $\hat{y}_v$ for $v \in \mathcal{T}_t$ on $G_t$ using $\theta_{t-1}$ (a forward pass of the GNN), and (ii) optionally update via a binary decision $\delta_t \in \{0, 1\}$. If $\delta_t = 1$, we fine-tune on the most recent $W$ steps of available labels:

$$\theta_t = \text{Train}\big(\theta_{t-1}, G_t, \mathcal{Y}_{[t-\tau-W+1:t-\tau]}\big),$$

and denote its training cost by $C_{\text{train},t}$. Otherwise $\theta_t = \theta_{t-1}$.

**Update Timing Problem.** Let $\ell(\theta_t; G_t, \mathcal{Y}_t)$ denote the task loss (e.g., cross-entropy) computed on the true labels for batch $t$, which become available only after a delay $\tau$. Given an initial model parameter $\theta_0$, the update timing problem aims to solve the following optimization problem:

$$\min_{\{\delta_t\}_{t=1}^T} \sum_{t=1}^T \ell(\theta_t; G_t, \mathcal{Y}_t) + \lambda \sum_{t=1}^T \delta_t C_{\text{train,t}},$$

$$\text{subject to} \quad \theta_t = \begin{cases} \theta_{t-1}, & \delta_t = 0, \\ \text{Train}(\theta_{t-1}, G_t, \mathcal{Y}_{[t-\tau-W+1:t-\tau]}), & \delta_t = 1, \end{cases} \tag{1}$$

where $\lambda > 0$ balances cumulative prediction loss against total training cost. Solving this requires predicting the task loss $\ell$ without access to the delayed labels $\mathcal{Y}_t$. The core of our approach is to use signals derived from the GNN's output embeddings as a proxy for this unobserved loss (detailed in §3.1), allowing us to make informed decisions about when to trigger an update.

## 3 THE GNNUPDATER FRAMEWORK

GNNUpdater is an adaptive framework that monitors a deployed GNN model and decides when to trigger an update. Its core objective is to balance model performance against computational cost (optimize Eq. 1). It consists of three main components: a Performance Predictor (§3.1), a Graph-Aware Update Trigger (§3.2), and a high-performance System Architecture (§3.3). A detailed illustration of the workflow is provided in Appendix A.

### 3.1 PERFORMANCE PREDICTOR

To address the challenge of predicting model performance before ground-truth labels are available, our goal is to learn a mapping

$$f : \mathbb{R}^k \to \mathbb{R}, \quad f(\mathbf{x}_{v,t}) \approx \text{model performance for node } v \text{ at time } t$$

by primarily leveraging shifts in node embeddings alongside other graph-aware features. This task is approached in two main stages:

**(1) Measuring Embedding Shifts (handle $p(x)$ drift).** We maintain two sets of embeddings for all relevant nodes: (i) *Reference embeddings* $\mathbf{H}_{\text{ref}}$, representing the node states captured by a full-graph inference performed immediately after the last model update. These serve as a stable baseline. (ii) *Current embeddings* $\mathbf{H}_t$, which are updated as new data batches arrive. Crucially, GNNUpdater is designed to be highly efficient by reusing these embeddings directly from the application's real-time inference pipeline, imposing negligible computational overhead. We denote a specific node $v$'s current and reference embedding as $\mathbf{h}_v^t \in \mathbf{H}_t$ and $\mathbf{h}_v^{\text{ref}} \in \mathbf{H}_{\text{ref}}$.[1]

---

[1] If node $v$ is a new node from $\Delta G_t$ and does not have its reference node embedding $\mathbf{h}_v^{\text{ref}}$, we can omit it during the computation, or use a dummy reference node embedding (e.g., all zeros).

We initially considered metrics such as:

$$\text{local\_drift}(v) = \|\mathbf{h}_v^t - \mathbf{h}_v^{\text{ref}}\|_1, \qquad \text{global\_drift} = \text{MMD}^2(\mathbf{H}_{\text{ref}}, \mathbf{H}_t),$$

where MMD refers to Maximum Mean Discrepancy Smola et al. (2006). Empirically, local drift alone can be overly sensitive to isolated, non-impactful changes, while global drift might average out or react too slowly to significant localized degradation. Consequently, neither metric by itself consistently and reliably tracks model performance across diverse datasets and GNN architectures (Table 1). Since GNNs derive node representations from their multi-hop neighborhoods, we assume that a node's behavior and its impact on downstream tasks are influenced not only by its own state but also by changes in its neighbors. This understanding motivates our aggregated *drift score*:

$$\text{drift}(v) = \frac{1}{|\{v\} \cup \mathcal{N}(v)|}\Big(\|\mathbf{h}_v^t - \mathbf{h}_v^{\text{ref}}\|_1 + \sum_{u \in \mathcal{N}(v)} \|\mathbf{h}_u^t - \mathbf{h}_u^{\text{ref}}\|_1\Big). \tag{2}$$

As shown in Figure 2, this aggregated score (Eq. 2), which averages the drift of node $v$ and its immediate neighbors $\mathcal{N}(v)$, exhibits a strong negative Pearson correlation (averaging –0.80 and up to –0.89) with downstream model performance. This significantly outperforms relying solely on local (average –0.60) or global (average –0.40) drift measures (Table 1), better reflecting the GNN's inherent neighborhood aggregation mechanism.

**(2) Learning the Mapping (handle $p(y|x)$ drift).** To predict performance, we construct a feature vector $\mathbf{x}_{v,t} \in \mathbb{R}^k$ for each target node $v$ at time $t$:

$$\mathbf{x}_{v,t} = \big[\, \text{drift}(v),\ \log(\text{num\_nodes}),\ \log(\text{num\_edges}),\ t,\ \deg(v)\,\big]. \tag{3}$$

We then collect training pairs $(\mathbf{x}_{v,t}, y_t)$, where $y_t$ is the actual realized performance (e.g., accuracy or NDCG Järvelin & Kekäläinen (2002)) obtained once labels are available. While simpler linear regression models can capture coarse performance trends, they often underfit and fail to model the complex, non-linear interactions between these features and the actual model performance. We found that a Random Forest regressor Breiman (2001) yields the best results. It effectively handles non-linear relationships and feature interactions, achieving prediction error as low as 3.5% and a Pearson correlation ($r$) with true performance up to 0.96 (details in Table 2).

Performance predictions are made in batches to reduce inference costs. The regressor is periodically retrained (a fast process, e.g., 0.25 seconds) as sufficient new labeled data accumulates. This ensures the predictor adapts to evolving mapping ($p(y|x)$) and maintains its accuracy over time. Figure 3 illustrates how our predictor, using this learned mapping, closely follows the actual performance, adeptly capturing both gradual decays and sudden improvements post-model-update.

### 3.2 GRAPH-AWARE UPDATE TRIGGER

The key idea of our graph-aware update trigger is to monitor model health on the *entire graph*, rather than just the most recently active target nodes. This global perspective enables us to detect performance degradation that is emerging in regions not currently being monitored. The detailed pseudo-code for the trigger algorithm can be found in Appendix B.

At each time $t$, we incorporate the latest batch of updates $\Delta G_t$ into our cumulative graph $G_{t-1}$ to form $G_t = G_{t-1} \cup \Delta G_t$. For each target node $v \in \mathcal{T}_t$, we construct the feature vector $\mathbf{x}_{v,t}$ (Eq. 3) and use our performance predictor (§3.1) to estimate its downstream-task performance $p_{v,t} = f(\mathbf{x}_{v,t})$, and classify it as either problematic or normal by comparing against a user-defined threshold $\varepsilon$:

$$y_v^{(0)} = \begin{cases} 1, & p_{v,t} < \varepsilon \quad \text{(problematic)}, \\ 0, & p_{v,t} \geq \varepsilon \quad \text{(normal)}, \end{cases}$$

with $y_u^{(0)} = 0$ for all non-target nodes $u \notin \mathcal{T}_t$.

We employ label propagation Zhu & Ghahramani (2002) to diffuse problematic labels through the graph structure to uncover hidden performance issues in regions not currently being monitored. This approach mirrors GNN neighborhood aggregation without requiring any learnable parameters.

Specifically, using the normalized adjacency matrix $\mathbf{S}_t = \mathbf{D}_t^{-1/2}\mathbf{A}_t\mathbf{D}_t^{-1/2}$ (where $\mathbf{D}_t$ and $\mathbf{A}_t$ are the degree matrix and adjacency matrix of $G_t$, respectively) and our initial labels $\mathbf{y}^{(0)} = [y_v^{(0)}]_{v \in V_t}$, we perform $k$ iterations:

$$\mathbf{y}^{(\ell+1)} = \alpha\, \mathbf{S}_t\, \mathbf{y}^{(\ell)} + (1 - \alpha)\, \mathbf{y}^{(0)}, \quad \ell = 0, \dots, k - 1.$$

where $\alpha$ controls the influence of neighboring nodes versus the initial classifications. In practice, only a few iterations (e.g., $k = 2$) suffice to detect early signs of global performance decay, making this process computationally efficient compared to a full GCN forward pass.

After propagation, each entry $y_v^{(k)}$ in the resulting vector reflects the degree to which node $v$ has been "influenced" by problematic nodes in its k-hop neighborhood. We then compute the global degradation ratio—the proportion of nodes across the entire graph that exceed our concern threshold:

$$r_t = \frac{1}{|V_t|} \sum_{v \in V_t} \mathbb{I}\big[y_v^{(k)} > 0.5\big].$$

This ratio serves as our primary signal for model health: if $r_t > \phi$, indicating widespread performance issues, we set $\delta_t = 1$ to trigger a model update and reset $\mathbf{y}^{(0)} \leftarrow \mathbf{0}$ to begin fresh monitoring; otherwise, we set $\delta_t = 0$ and continue with the current model.

By unifying direct performance estimates on recent target nodes with this graph-aware propagation process, our mechanism effectively captures both immediate failures and subtle multi-hop drift patterns that traditional approaches would miss. This ensures updates are triggered precisely when needed—when degradation has truly become a global rather than merely local phenomenon.

**Justification for Threshold-Based Trigger.** The use of thresholds ($\varepsilon$ and $\phi$) is a deliberate design choice to provide a simple, interpretable, and computationally efficient mechanism for controlling the update policy. These thresholds serve as direct levers for system operators to balance performance against computational cost. The node-level performance threshold, $\varepsilon$, can be set based on the minimum acceptable performance required by the specific application (e.g., an SLA for prediction accuracy). The global trigger threshold, $\phi$, can be tuned on a validation set of historical data to find an optimal point on the performance-cost curve, which effectively solves the optimization problem from Section 2. Varying $\phi$ allows operators to approximate different settings of the cost-balancing hyperparameter $\lambda$, making the system adaptable to diverse operational constraints.

### 3.3 GNNUpdater System Architecture

GNNUpdater is a high-performance distributed library built on PyTorch and DGL, designed for efficient learning on billion-edge dynamic graphs. The performance prediction module is built with scikit-learn, while the graph-aware update trigger relies on PyTorch and torch_sparse Fey (2024).

A key design focus of GNNUpdater is the efficient processing of large-scale dynamic graphs. Most existing GNN frameworks are optimized for static graphs; for example, widely used systems like DGL Wang et al. (2019) often require rebuilding or re-partitioning the graph structure when changes occur, causing significant computational overhead. To address these limitations, GNNUpdater employs a specialized C++ block-based streaming graph storage system, enabling efficient incremental updates. This is further complemented by a CUDA-accelerated GPU neighbor finder for fast neighborhood sampling and a PyTorch-based dynamic GPU feature cache for low-latency feature access.

To scale to billion-edge graphs, GNNUpdater supports distributed training through graph partitioning and leverages PyTorch's asynchronous RPC framework. Additional details about the system architecture are provided in Appendix C.

## 4 Related Works

**Continual Graph Learning.** Continual graph learning (CGL) aims to learn incrementally on evolving graphs without catastrophic forgetting Xu et al. (2020b); Wang et al. (2021b; 2020); Perini et al. (2022); Ahrabian et al. (2021); Ding et al. (2022); Yuan et al. (2023); Su et al. (2023). Existing works primarily focus on mitigating forgetting when updating models Yuan et al. (2023), using techniques like parameter importance scoring Liu et al. (2021), knowledge distillation Xu et al. (2020b), experience replay Wang et al. (2020), or regularization Su et al. (2023). Unlike these works, which focus on *how* to update, GNNUpdater addresses the under-explored problem of *when* to update, making it a complementary component to existing CGL methods.

**Learning on Dynamic Graphs.** Prior work has developed GNNs for dynamic or temporal graphs Trivedi et al. (2019); Pareja et al. (2020); Ma et al. (2020); Sankar et al. (2020); Wang et al. (2021a). Examples include TGAT Xu et al. (2020a), which uses self-attention to aggregate temporal-topological features, TGN Rossi et al. (2021), maintaining node states via a memory module, and ROLAND You et al. (2022), offering generic solutions for dynamic GNNs. These works

focus on *how* to learn from evolving structures, while our work addresses the orthogonal question of *when* to update them—making GNNUpdater a complementary support layer for dynamic GNNs.

**Concept Drift Detection.** Detecting concept drift is a well-studied problem in traditional data streams Lu et al. (2018), with methods typically divided into two families. **Data-based detectors** monitor input distributions but are generally unsuitable for graphs as they cannot account for topological structure Gama et al. (2014b). More common are **performance-based detectors**, which monitor a stream of metrics like error rate. Some, like DDM Gama et al. (2004), track the online error rate and trigger when it exceeds a threshold, while others, like KSWIN Raab et al. (2020), apply statistical tests to detect changes in the stream. These methods, however, are ill-suited for many graph applications due to two key challenges: (1) **label delay**, as they require fresh ground-truth labels to compute performance, and (2) **hidden drift**, as they monitor an aggregated stream and cannot detect degradation from local structural changes. GNNUpdater is designed to overcome both limitations.

## 5 EVALUATION

In this section, we focus on evaluating the effectiveness of GNNUpdater's adaptive update strategy against various baselines. Detailed evaluations of the system's performance (§3.3) against DGL, including graph operation costs and scalability, are presented in Appendix E. Our system optimizations yield substantial improvements, reducing graph operation overhead by up to 92.5% and demonstrating near-linear scaling on multi-GPU setups for billion-edge graphs. All experiments use our optimized system implementation for fair comparison.

**Datasets and Models.** We evaluate GNNUpdater on three datasets from the Temporal Graph Benchmark (TGB)Huang et al. (2023): `tgbn-genre`Kumar et al. (2019), `tgbn-reddit`Nadiri & Takes (2022), and `tgbn-token`Shamsi et al. (2022). We adopt the node affinity prediction task Huang et al. (2023), which forecasts users' preference scores over all items for the upcoming 7 days. Temporal events are aggregated daily, and predictions are evaluated using NDCG@10 Wu et al. (2022), a standard metric for top-k recommendations comparing ranked recommendations against actual user interactions during the prediction period. Due to the forward-looking nature of predictions, ground truth labels are available only after a 7-day delay. This delay is characteristic of real-world applications like recommendation systems (where user feedback accrues over time) or fraud detection (where labels require manual investigation). We use three representative GNN models: GraphSAGE Hamilton et al. (2017), GAT Veličković et al. (2018), and TGAT Xu et al. (2020a), all with 2-layer architectures, 100-dimensional embeddings, and sampling 10 neighbors per node per layer. Each is combined with a simple MLP decoder mapping node embeddings to preference scores. Detailed dataset characteristics and model configurations are in Appendix D.1.

**Methodology.** Our experiments simulate a continuous data generation scenario where new data is fed to the system on a daily basis for model updating. A base model is initially trained on the first 30% of the total timespan of the dataset. Thereafter, daily aggregated data is sent to GNNUpdater, and the system determines whether to trigger fine-tuning based on the specified update trigger method. Upon triggering, and in line with common practice Tian et al. (2018); Yuan et al. (2023); Wang et al. (2024), each fine-tuning operation uses the past 365 days of labeled data (offset by 7 days due to the label lag) and is performed for one epoch to balance stability and plasticity. Detailed training configurations are in Appendix D.2.

**Baselines.** We compare GNNUpdater to the following baselines (DTP = delayed true performance, PP = predicted performance): (1) **Periodic Update:** Fixed-time interval updates regardless of performance; (2) **PerfDrop-DTP:** Triggers update when 7-day mean NDCG@10 (with 7-day label delay) drops below the predefined threshold $\epsilon$; (3) **PerfDrop-PP:** Triggers update when 7-day mean predicted NDCG@10 drops below the predefined threshold $\epsilon$; (4) **ADWIN-DTP:** ADWIN drift detector Bifet & Gavalda (2007) on DTP stream; (5) **ADWIN-PP:** ADWIN applied to PP stream with identical configuration; (6) **KSWIN-DTP:** Kolmogorov-Smirnov Windowing (KSWIN) drift detector Raab et al. (2020) on DTP stream; (7) **KSWIN-PP:** KSWIN applied to PP stream with identical parameters; (8) **MMD:** Triggers update when the Maximum Mean Discrepancy (MMD) Smola et al. (2006) score exceeds a specified threshold. Detailed method settings and GNNUpdater's sensitivity analysis can be found in Appendix D.3.

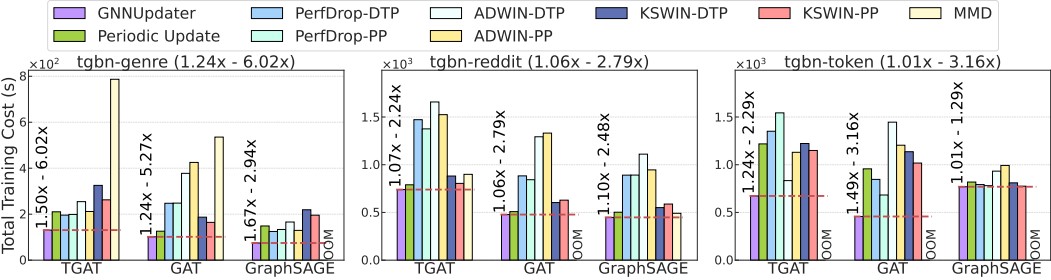

(a) Average performance comparison under equal total training cost constraints across various datasets and models. GNNUpdater improves performance by 5.3% on average.

(b) Total training cost comparison when maintaining equivalent average performance (absolute difference < 0.001) across various datasets and models. GNNUpdater reduces training costs by 2.0× on average.

Figure 4: GNNUpdater improves performance at equal training cost or matches their performance with far less training effort. OOM indicates GPU out-of-memory error.

## 5.1 MAIN RESULTS

**Evaluation Metrics.** We evaluate GNNUpdater using two primary metrics: (1) average downstream task performance (NDCG@10) across streaming batches, and (2) total training cost incurred due to model updates. Our analysis is twofold: first, we compare the average performance achieved by different methods under an identical total training cost constraint; second, we compare the total training cost required by each method to maintain an equivalent level of average performance (defined as an absolute NDCG@10 difference of less than 0.001). All methods were optimized via extensive hyperparameter grid search for each scenario (see Appendix D.4 for tuning strategy and parameters), ensuring GNNUpdater is benchmarked against strong baseline configurations.

**Superior Performance Under Fixed Training Budget.** Figure 4a compares the average performance when total training costs are equalized across all methods. GNNUpdater consistently achieves superior accuracy, improving the average NDCG@10 by **5.3% on average** across all datasets and models. The maximum observed improvement reaches **34.0%** (specifically, for GAT on `tgbn-token` compared to ADWIN-DTP). Such gains are significant in production, where even minor accuracy improvements can yield substantial impacts Lin et al. (2022); Tian et al. (2018).

The improvements are most pronounced on `tgbn-token` (up to 34.0%), likely due to its complex transaction patterns. Cryptocurrency transaction networks evolve rapidly, causing models to quickly become outdated Shamsi et al. (2022). `tgbn-reddit` shows more moderate gains (up to 2.9%), reflecting its relatively stable social network structure.

While various baselines perform competitively on specific datasets (e.g., PerfDrop variants on `tgbn-genre`, periodic updates and MMD on `tgbn-reddit`, KSWIN variants on `tgbn-token`), they consistently trail GNNUpdater. Notably, MMD, despite potential competitiveness, frequently encounters out-of-memory (OOM) errors on large datasets due to its quadratic complexity, limiting its practical use. These results underscore the consistent effectiveness of GNNUpdater's update strategy.

**Reduced Training Cost for Equivalent Performance.** Figure 4b illustrates the total training cost required for each method to match the average performance achieved by GNNUpdater. Our approach reduces training costs by a factor of **2.0× on average**. The maximum cost reduction observed is **6.0×** (for TGAT on `tgbn-genre` compared to MMD). The efficiency gains are particularly evident on `tgbn-genre`, where the best-performing baselines still necessitate 19-40% more training time. MMD's high cost on this dataset is exacerbated by frequent false positives that trigger unnecessary updates. On `tgbn-reddit`, periodic updates are relatively cost-effective but still incur 6-9% more

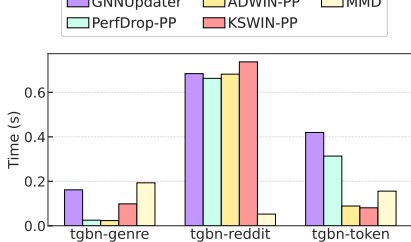

Figure 5: Performance comparison of GNNUpdater variants under equal cost and matched performance scenarios.

Figure 6: Comparison of per-batch overhead (in seconds) of different methods.

overhead than GNNUpdater. For `tgbn-token`, the overhead of baselines varies (from 0.78% for KSWIN-DTP with GraphSAGE to 33% for PerfDrop-PP with GAT). These findings highlight the significant training efficiency benefits provided by GNNUpdater across diverse datasets.

## 5.2 Ablation Studies

**Performance Predictor.** We first evaluate the performance predictor detailed in §3.1. Table 2 demonstrates its high accuracy, with error rates as low as 3.51% and correlation up to 0.96. This precision is attributed to two main factors: (1) the aggregated drift score (Eq. 2) effectively captures both local and neighborhood embedding shifts, and (2) the Random Forest regressor successfully models the complex non-linear relationships between these drift patterns and performance degradation.

To validate our design, we first compare different embedding difference metrics (Table 1). The correlation results are averaged from all datasets and models. Simple node-level metrics like L1/L2 distances show moderate correlation (-0.60) but high variance, indicating they miss important structural changes. Distribution-based metrics like MMD and Central Moment Discrepancy (CMD) Zellinger et al. (2017) are computationally expensive (850ms and 160ms, respectively) while achieving lower or unstable correlations. Our 1-hop neighbor L1 distance achieves both the strongest (-0.80) and most stable (±0.06) correlation while maintaining reasonable computation time (70ms). While 2-hop

Table 1: Comparison of different embedding difference metrics.

| Metric Type | Correlation with NDCG@10 | Compute Time (ms) |
|---|---|---|
| *Node-level Metrics* | | |
| L1 distance | -0.60±0.25 | 20 |
| L2 distance | -0.57±0.25 | 20 |
| *Distribution-based Metrics* | | |
| MMD | -0.40±0.46 | 850 |
| CMD | -0.67±0.30 | 160 |
| *Neighbor-aware Metrics* | | |
| 1-hop Neighbor L1 Distance | **-0.80±0.06** | **70** |
| 2-hop Neighbor L1 Distance | -0.81±0.10 | 240 |

neighborhoods marginally improve correlation (-0.81), the increased standard deviation (±0.10) and computation time (240ms) suggest that it introduces noise while being less efficient. These results validate our choice of 1-hop neighbor aggregation in our drift score design (Eq. 2), which provides the optimal balance between prediction accuracy and computational efficiency.

Next, we compare against linear regression (LR) as an alternative predictor. Table 2 shows that Random Forest (RF) consistently outperforms LR across all datasets and GNN architectures. On the complex `tgbn-token` dataset, RF reduces prediction error by over 27% relative to LR. RF still achieves meaningful improvements even on stable datasets like `tgbn-reddit` where LR performs well (correlation > 0.90). These results confirm that capturing non-linear relationships between our features and model performance is crucial for accurate prediction.

Table 2: Performance prediction error and Pearson correlation comparison between Linear Regression (LR) and Random Forest (RF) regressors.

| Dataset | Model | Linear Regression | | Random Forest | |
|---|---|---|---|---|---|
| | | Error(%) | Corr | Error(%) | Corr |
| `tgbn-genre` | TGAT | 5.93 | 0.60 | **5.14** | **0.69** |
| | GAT | 5.94 | 0.57 | **5.51** | **0.63** |
| | GraphSAGE | 5.66 | 0.41 | **5.12** | **0.52** |
| `tgbn-reddit` | TGAT | 5.21 | 0.91 | **3.97** | **0.94** |
| | GAT | 4.75 | 0.91 | **3.87** | **0.94** |
| | GraphSAGE | 4.65 | 0.90 | **3.51** | **0.93** |
| `tgbn-token` | TGAT | 15.91 | 0.92 | **12.05** | **0.96** |
| | GAT | 18.62 | 0.94 | **13.48** | **0.96** |
| | GraphSAGE | 16.06 | 0.90 | **13.33** | **0.95** |

**Graph-aware Trigger.** Next, we examine the efficacy of our graph-aware update trigger (§3.2) using GAT by comparing three variants: (1) our full design using predicted performance with label propagation (GNNUpdater-PP), (2) a version that omits the label propagation step and relies solely

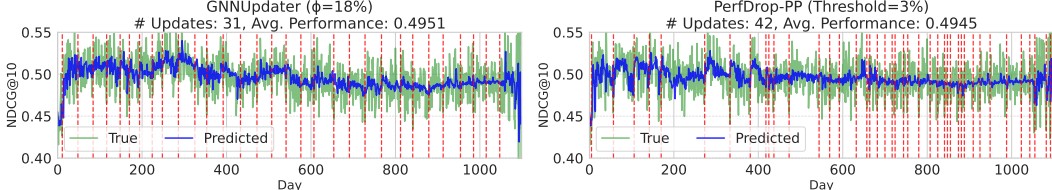

Figure 7: Update trigger patterns comparison between GNNUpdater (left) and PerfDrop-PP (right). Blue/green lines show predicted/true NDCG@10 scores; red dashed lines indicate model updates. GNNUpdater achieves similar performance with fewer, more evenly distributed updates.

on performance prediction from the current batch (GNNUpdater (no propagation)), and (3) a baseline that triggers updates based on delayed true performance (GNNUpdater-DTP). Similar to §5.1, we examine two scenarios, as shown in Figure 5.

Under equal computational cost, our full approach achieves up to 1.81× higher NDCG@10 across datasets. The largest gains are observed on `tgbn-token` compared to variants without label propagation, where complex transaction patterns make hidden drift detection crucial. When targeting matched average performance (absolute difference < 0.001), variants without label propagation require up to 1.84× more training costs. Similarly, GNNUpdater-DTP, using delayed true performance, suffers from label delay, resulting in higher training costs (up to 1.48×) despite similar performance. These results demonstrate that label propagation combined with performance prediction helps evaluate the global model health more accurately, thus making better trigger timing decisions.

**Trigger Update Algorithm's Overhead.** Figure 6 compares the trigger algorithm overhead per batch between GNNUpdater and state-of-the-art update trigger methods. We exclude simple methods like periodic updates from this comparison as they incur negligible computational overhead. MMD's reported overhead is measured before encountering out-of-memory errors. Our method incurs comparable computational costs with most baseline methods, requiring about **0.6 seconds** for each update. The low overhead is attributed to our lightweight performance predictor design and efficient label propagation implementation.

## 5.3 CASE STUDY

To better understand the advantages of our graph-aware update trigger, we present a detailed case study comparing GNNUpdater with the traditional PerfDrop-PP trigger on the `tgbn-genre` dataset using the GAT model. Figure 7 compares trigger patterns between PerfDrop-PP and GNNUpdater's graph-aware approach. While achieving a similar NDCG@10 (0.4951 vs 0.4945), GNNUpdater requires fewer updates (31 vs 42). GNNUpdater's update trigger demonstrates a more uniform distribution, while PerfDrop-PP shows a clustered pattern. For example, PerfDrop-PP triggers multiple updates in quick succession during days 400-450, 600-800, 800-900, and 1000-1100, yet these clustered updates yield minimal performance improvements. In contrast, GNNUpdater uses graph awareness to assess if performance degradation is widespread and significant before triggering. This structural insight prevents the frequent, less effective updates seen with PerfDrop-PP, leading to more efficient model maintenance with fewer, yet more impactful, updates while preserving performance.

## 6 CONCLUSION

We propose GNNUpdater, a high-performance distributed system that addresses the critical challenge of determining when to update GNN models in continual graph learning. By combining a lightweight performance predictor (correlating embedding drift with downstream accuracy) and graph-aware label propagation (evaluating model's global effectiveness on the whole graph) GNNUpdater achieves significant improvements: 5.3% higher accuracy on average (up to 34.0%) under fixed training budgets and 2.0× lower training costs at matched accuracy compared to various baselines.

**Limitation and Future Work.** Our empirical evaluation, while comprehensive, is focused on the task of node affinity prediction on homogeneous graphs. A key direction for future work is to validate the broader applicability of GNNUpdater by extending it to other common dynamic graph learning tasks, such as dynamic node classification and link prediction, and to the important domain of heterogeneous graphs Gastinger et al. (2024). We believe our core principles of a label-delay tolerant performance predictor and a graph-aware trigger provide a strong foundation for these extensions.

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

# A  OVERVIEW

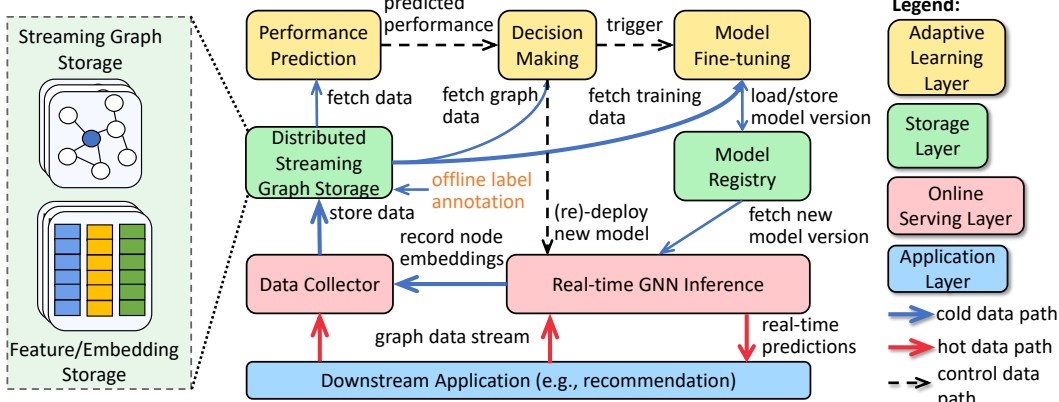

Figure 8: Overview of GNNUpdater.

GNNUpdater operates as an adaptive framework that monitors a deployed GNN model and intelligently decides when to trigger an update. An overview of the architecture can be found in Figure 8. The core workflow is as follows: GNNUpdater operates as follows:

1. **Streaming Graph Ingestion**: The system ingests graph updates through a block-based storage structure that organizes temporal edges as linked lists of blocks of edges. This enables efficient appends and neighborhood queries without full graph reconstruction (§3.3).

2. **Inference and Embedding Generation**: The deployed GNN model runs inference, generating new node embeddings for nodes in the graph updates. These embeddings are asynchronously stored in the feature/embedding storage and update the current set of node embeddings.

3. **Performance Prediction via Embedding Shift Monitoring**: For target nodes in the graph updates (i.e., nodes we aim to make predictions), the system collects the latest node embedding and the embeddings of its neighbors from the current embedding set. The aggregated drift scores are then calculated between the current embeddings and the corresponding reference embeddings (generated after the last model update). A lightweight Random Forest regressor takes these aggregated drift scores along with temporal graph statistics (e.g., node/edge growth rates, degree distribution) as input to predict performance. The regressor will be retrained periodically when enough labeled data arrive (§3.1).

4. **Graph-Aware Decision Making**: The system continuously predicts performance for incoming target nodes and classifies nodes as problematic when the predicted performance is below a threshold. Label propagation is applied to propagate these labels (problematic or normal) from already classified nodes to their neighbors in the graph. If the number of problematic nodes exceeds a capacity threshold, a model fine-tuning is immediately triggered (§3.2).

5. **Model Update**: The system fine-tunes the model using a sliding window of available labeled data and obtains the updated GNN model, which is stored in the model registry. The online serving module then fetches and deploys the updated model. Finally, the reference node embeddings are re-generated through a full-graph inference.

# B  GRAPH-AWARE UPDATE TRIGGER ALGORITHM

# C  SYSTEM DESIGN AND IMPLEMENTATION

GNNUpdater is implemented with 13.5k lines of code (LoC) and incorporates efficient techniques for handling dynamic graphs at scale. This section describes our implementation of core components and key optimizations.

---

**Algorithm 1** Graph-Aware Update Trigger Algorithm

---

**Require:** Current graph $G_{t-1} = (V_{t-1}, E_{t-1})$, incoming stream of edge-update batches $\{\Delta G_t\}_{t=1}^{T}$
**Require:** Target nodes $\mathcal{T}_t \subseteq V_t$ for each batch $t$
**Require:** Propagation weight $\alpha \in (0, 1)$, propagation iterations $k$
**Require:** Problematic-node ratio threshold $\phi \in (0, 1)$, performance threshold $\varepsilon$
**Ensure:** Decision $\delta_t \in \{\text{trigger}, \text{no-trigger}\}$ each batch
 1: Initialize predicted-label vector $\mathbf{y} \in \{0, 1\}^{|V|}$ to all zeros
 2: **while** new batch $\Delta G_t$ arrives **do**
 3:    $G_t \leftarrow G_{t-1} \cup \Delta G_t$     ▷ update cumulative graph
 4:    $\mathbf{p}_t \leftarrow \text{PredictPerf}(\mathcal{T}_t)$     ▷ performance estimate for each $v \in \mathcal{T}_t$
 5:    **for all** $v \in \mathcal{T}_t$ **do**
 6:        **if** $p_{v,t} < \varepsilon$ **then**
 7:            $y_v \leftarrow 1$     ▷ mark as problematic
 8:        **else**
 9:            $y_v \leftarrow 0$     ▷ mark as normal
10:        **end if**
11:    **end for**
12:    $\mathbf{S}_t \leftarrow \mathbf{D}_t^{-1/2} \mathbf{A}_t \mathbf{D}_t^{-1/2}$     ▷ normalized adjacency of $G_t$
13:    $\mathbf{y}^{(0)} \leftarrow \mathbf{y}$
14:    **for** $i = 1 \dots k$ **do**
15:        $\mathbf{y} \leftarrow \alpha \mathbf{S}_t \mathbf{y} + (1 - \alpha) \mathbf{y}^{(0)}$     ▷ propagate problematic scores
16:    **end for**
17:    $r \leftarrow \left| \{ v \in V_t | y_v > 0.5 \} \right| / |V_t|$
18:    **if** $r > \phi$ **then**
19:        **trigger** model update ($\delta_t = \text{trigger}$)
20:        $\mathbf{y} \leftarrow \mathbf{0}$     ▷ reset for next batch
21:    **else**
22:        **no-trigger** ($\delta_t = \text{no-trigger}$)
23:    **end if**
24:    $t \leftarrow t + 1$
25: **end while**

---

## C.1   CORE COMPONENTS FOR UPDATE TIMING DECISIONS

**Performance Prediction Module** (§3.1, 500 LoC) is implemented in Python and maps embedding drift features to downstream performance estimates. We use a Random Forest regressor from `scikit-learn` 1.5.2 Pedregosa et al. (2011). The GNN models leverage PyTorch 2.4.0 (CUDA 12.4) and DGL 2.4.0 Wang et al. (2019) to generate embeddings used in drift computation. Reference and current node embeddings are stored in a Python dictionary mapping node IDs to embedding tensors.

**Graph-Aware Update Trigger** (700 LoC) is implemented in Python using PyTorch and the `torch_sparse` package Fey (2024) for efficient sparse matrix operations. Our implementation leverages PyG 2.6.1's Fey & Lenssen (2019) label propagation primitive and supports dynamic resizing of propagation matrices for incremental processing of large-scale graphs.

## C.2   STREAMING GRAPH INFRASTRUCTURE

To efficiently support graph updates and queries at scale, GNNUpdater introduces three key system optimizations:

**Block-Based Streaming Graph Storage** (2,200 LoC, C++). Unlike traditional graph frameworks (e.g., DGL, PyG Fey & Lenssen (2019)) that require full graph reconstruction during updates, GNNUpdater implements a block-based storage structure optimized for streaming graphs (Figure 9). Each node's outgoing edges are stored as a doubly linked list of chronologically ordered lightweight blocks containing: edge data pointers, block size (typically 64-256 edges), minimum/maximum timestamps, and previous/next block pointers (72 bytes total per block).

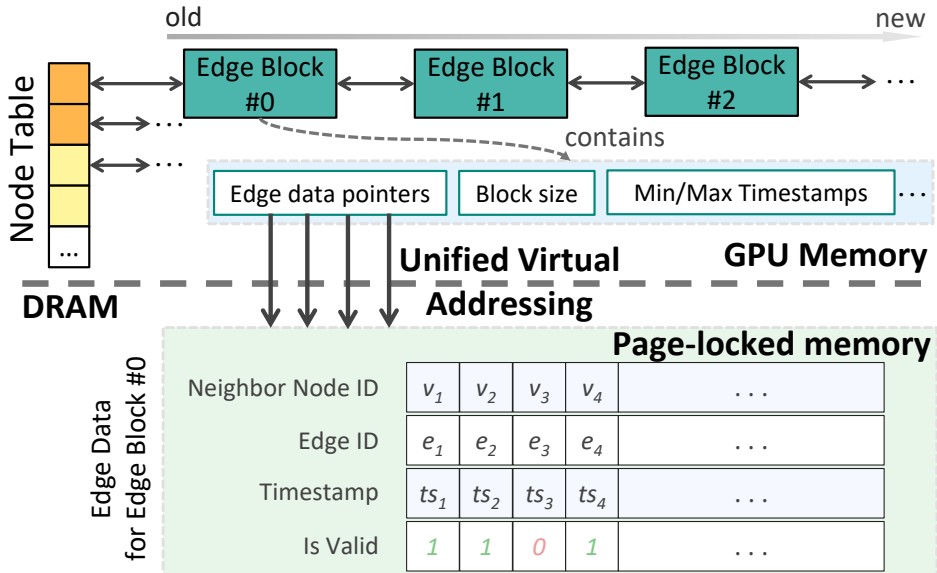

Figure 9: Block-based streaming graph storage architecture. Each node maintains a linked list of edge blocks containing metadata (block size, timestamps, pointers) and references to edge data arrays. Lightweight metadata (72 bytes/block) resides in GPU to enable efficient neighbor finding, while heavy edge data resides in page-locked host memory for scalability.

Edge data arrays store neighbor IDs, edge IDs, timestamps, and validity flags in page-locked host memory. New edges are appended to existing blocks until capacity is reached, triggering new block allocation. The chronological ordering enables $O(1)$ appends and efficient temporal queries via timestamp filtering. The node table is implemented using `thrust` package's NVIDIA (2024) device_vector and page-locked host memory is allocated via `rmm` AI (2024).

**GPU-Based Neighbor Finder** (400 LoC, CUDA). Generating node embeddings and calculating drift scores (Eq.2) require efficient neighbor queries. Our approach keeps critical metadata (node table and block lists) in GPU memory (<100MB typically) while maintaining edge data (potentially several GBs) in pinned host memory for scalability. The Neighbor Finder leverages Unified Virtual Addressing (UVA) to access host-based edge data directly from GPU kernels.

Our custom CUDA kernels optimize performance by launching one GPU thread per neighbor sample, with adjacent threads working on the same target node to reduce warp divergence. Candidate neighbor positions are cached in GPU shared memory for uniform sampling, and sampled results are written consecutively to global memory to support coalesced access.

**GPU Feature Cache** (PyTorch). Raw node/edge features critical for computing embeddings are often too large to fully reside in GPU memory Yuan et al. (2024). GNNUpdater employs an LRU cache that prioritizes frequently accessed data in GPU memory while maintaining less-used features in host memory, fetched on-demand via UVA. This strategy reduces feature retrieval latency by 58% compared to CPU-only access in our experiments. The cache maintains "recent scores" for entries in a tensor, with the lowest-scoring entries efficiently evicted via vectorized top-k operations.

## C.3 DISTRIBUTED TRAINING

GNNUpdater supports distributed training for large-scale graphs through graph partitioning using various methods (e.g., hash, LDG Stanton & Kliot (2012)). For distributed operations, we employ edge-cut partitioning and PyTorch's asynchronous RPC framework to handle remote calls for edge additions, neighbor finding, and feature fetching.

To ensure balanced workloads, when a trainer needs to find a remote target node's neighbors, it sends an RPC request to the machine hosting that node, targeting a GPU that mirrors the trainer's local GPU rank. The distributed feature/embedding store is implemented using Python dictionaries with

PyTorch's RPC for data transmission. This architecture enables GNNUpdater to handle billion-edge graphs, as demonstrated in our evaluation (Appendix §E).

# D EXPERIMENTAL DETAILS

## D.1 DATASETS AND MODELS

Table 3: Dataset Statistics. * denotes randomized features. $|d_v|$ and $|d_e|$ show the dimensions of node features and edge features, respectively. The first 30% of the temporal data is used as the initial training period.

|  | tgbn-genre | tgbn-reddit | tgbn-token |
| --- | --- | --- | --- |
| Domain | Recommendation | Social Network | Transaction Network |
| $|V|$ | 1,505 | 11,766 | 61,756 |
| $|E|$ | 17.9M | 27.2M | 72.9M |
| Duration (month) | 52 | 36 | 26 |
| #Batches | 1,580 | 1,090 | 785 |
| $|d_v|$ | 100* | 100* | 100 |
| $|d_e|$ | 1 | 1 | 2 |

**Testbed.** Unless otherwise specified, we conduct all experiments on two machines, each equipped with 4 x 40 GB A100 GPUs, 256 AMD EPYC 7H12 64-Core processors, 512 GB DRAM, and a 100 Gbps Ethernet network.

**Datasets.** We evaluate GNNUpdater on three real-world datasets from the Temporal Graph Benchmark (TGB) Huang et al. (2023): tgbn-genre Kumar et al. (2019), tgbn-reddit Nadiri & Takes (2022), and tgbn-token Shamsi et al. (2022). Detailed statistics for each dataset are given in Table 3. tgbn-genre is a bipartite and weighted interaction network between users and the music genres of songs they listen to Kumar et al. (2019). tgbn-reddit is a users and subreddits interaction network Nadiri & Takes (2022). tgbn-token is a user and cryptocurrency token transaction network Shamsi et al. (2022).

**Models.** We use three representative GNN models in our experiments: GraphSAGE Hamilton et al. (2017), GAT Veličković et al. (2018), and TGAT Xu et al. (2020a). Following their original implementations, all models use a 2-layer architecture where each layer randomly samples 10 neighbors per node, with 100-dimensional hidden embeddings. For all models, we set dropout to 0.1. For GraphSAGE, we use the mean aggregator. And for GAT and TGAT, we use 2 attention heads. For TGAT, the temporal encoding dimension is 100.

## D.2 TRAINING CONFIGURATIONS

For the initial base model training with the first 30% of data, we train each base model for 200 epochs with early stopping, using the Adam optimizer Kingma & Ba (2015) with a learning rate of $1 \times 10^{-3}$. For each fine-tuning operation triggered by any method, we use the past 365 days of labeled data (offset by 7 days due to the inherent label delay in the task) and fine-tune the GNN model for one epoch. The Adam optimizer is used with a fine-tuning learning rate of $1 \times 10^{-4}$. No weight decay is applied during fine-tuning. We use a batch size of 2,000 for tgbn-genre and tgbn-token, and 4,000 for tgbn-reddit, as it is larger.

## D.3 METHOD SETTINGS AND SENSITIVITY ANALYSIS

For ADWIN and KSWIN, we adopt the scikit-multiflow package Montiel et al. (2018) and adjust the default window size to 15 for a fair comparison with other methods. For MMD implementation, we use the alibi-detect package Van Looveren et al. (2019) and leverage PyTorch with CUDA acceleration as the backend, since the CPU version is prohibitively slow. We set the performance threshold $\epsilon$ to 0.50 for both tgbn-genre and tgbn-reddit, and 0.25 for tgbn-token, based on their initial base model training performance and the recommendations in the original paper Huang et al. (2023).

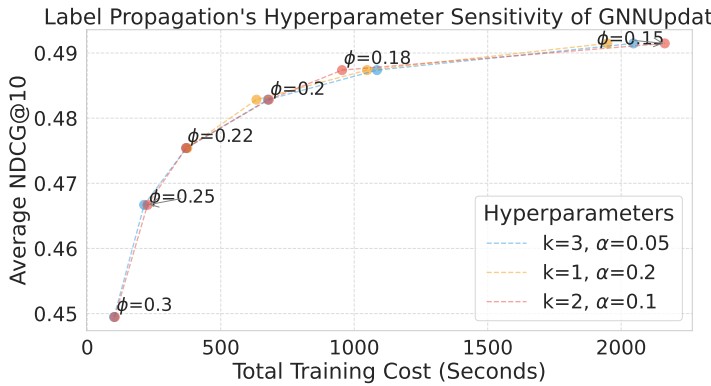

Figure 10: Impact of label propagation hyperparameters on GNNUpdater's performance-cost trade-off, using the TGAT model on the `tgbn-genre` dataset. The close similarity across different settings indicates GNNUpdater's robustness to these specific label propagation hyperparameter choices.

By default, GNNUpdater combines performance predictor (§3.1) with 2-layer label propagation (§3.2) with propagation layers $k = 2$ and propagation weight $\alpha = 0.1$. We also investigated the sensitivity of GNNUpdater to the choice of its label propagation hyperparameters: the number of propagation layers $k$ and the propagation weight $\alpha$. Figure 10 presents this analysis for the TGAT model on the `tgbn-genre` dataset. Other models and datasets show similar results. It plots the Average NDCG@10 against Total Training Cost for three distinct $(k, \alpha)$ pairs—specifically $(k = 3, \alpha = 0.05)$, $(k = 1, \alpha = 0.2)$, and our default setting of $(k = 2, \alpha = 0.1)$—across a range of problematic node ratio thresholds ($\phi$). The results demonstrate that all three configurations yield remarkably similar performance-cost trade-off curves. This low sensitivity indicates that GNNUpdater's effectiveness is robust to reasonable variations in these label propagation parameters.

### D.4 Hyperparameter Optimization Strategy

To ensure a rigorous and fair comparison for each dataset and GNN model combination, we performed an extensive grid search over the hyperparameters for GNNUpdater and all baseline methods. For each of the two experimental scenarios described in §5.1, the reported results for every method (including GNNUpdater) correspond to the hyperparameter configuration that optimized its outcome under that specific scenario's objective (i.e., maximizing NDCG@10 for a fixed cost, or minimizing cost for a target NDCG@10). This systematic tuning ensures that GNNUpdater is benchmarked against the most competitive configurations of all baselines. The searched hyperparameters and their respective value ranges for each dataset were as follows:

#### D.4.1 `TGBN-GENRE`

- PerfDrop Variants (%): `[0.005, 0.01, 0.02, 0.03, 0.04, 0.05, 0.07, 0.08, 0.1, 0.15]`
- Periodic Update (Intervals in days): `[14, 30, 60, 90, 180]`
- GNNUpdater (Problematic Node Ratio $\phi$): `[0.15, 0.18, 0.2, 0.22, 0.25, 0.3]`
- ADWIN (Delta $\delta$): `[1.0, 2.0, 5.0, 5.5, 6.0, 6.5, 7.0, 10.0]`
- KSWIN (Significance Level $\alpha_{\text{KSWIN}}$): `[0.001, 0.005, 0.01, 0.05, 0.1, 0.3, 0.5, 0.7, 0.8, 0.9]`
- MMD (Distance Threshold): `[-0.0003, -0.0005, -0.0007, -0.0008, -0.0009, -0.001, -0.0012]`

#### D.4.2 `TGBN-REDDIT`

- PerfDrop Variants (%): `[0.01, 0.02, 0.03, 0.04, 0.05, 0.1]`

- Periodic Update (Intervals in days): `[7, 14, 30, 60, 90]`
- GNNUpdater (Problematic Node Ratio $\phi$): `[0.3, 0.4, 0.42, 0.45, 0.47, 0.5]`
- ADWIN (Delta $\delta$): `[1.0, 2.0, 5.0, 5.5, 6.0, 6.5, 7.0, 10.0]`
- KSWIN (Significance Level $\alpha_{\text{KSWIN}}$): `[0.001, 0.005, 0.01, 0.05, 0.1, 0.3, 0.5, 0.7, 0.8, 0.9]`
- MMD (Distance Threshold): `[0.0015, 0.0012, 0.001, 0.0007, 0.0005, 0.0003]`

### D.4.3  `TGBN-TOKEN`

- PerfDrop Variants (%): `[0.1, 0.2, 0.3, 0.4, 0.5]`
- Periodic Update (Intervals in days): `[7, 14, 30, 60, 90]`
- GNNUpdater (Problematic Node Ratio $\phi$): `[0.3, 0.4, 0.45, 0.5, 0.55, 0.56, 0.57, 0.58, 0.6, 0.7]`
- ADWIN (Delta $\delta$): `[1.0, 2.0, 5.0, 5.5, 6.0, 6.5, 7.0, 10.0]`
- KSWIN (Significance Level $\alpha_{\text{KSWIN}}$): `[0.008, 0.01, 0.05, 0.1, 0.3, 0.5, 0.7, 0.8, 0.9]`
- MMD (Distance Threshold): `[0.2, 0.15, 0.13, 0.1, 0.08, 0.07, 0.05]`

Table 4: Graph operation overhead with DGL across all datasets (in milliseconds per batch)

| Dataset | Component | GNNUpdater | DGL |
|---|---|---|---|
| `tgbn-genre` | Graph Updates | **20.88** | 118.58 |
| | Neighbor Finding | **2.65** | 3.20 |
| | Feature Loading | **1.76** | 214.55 |
| | Total | **25.29** | 336.33 |
| `tgbn-reddit` | Graph Updates | **52.32** | 130.26 |
| | Neighbor Finding | **10.74** | 12.31 |
| | Feature Loading | **10.70** | 235.30 |
| | Total | **75.76** | 377.87 |
| `tgbn-token` | Graph Updates | **9.09** | 81.03 |
| | Neighbor Finding | 4.48 | **3.07** |
| | Feature Loading | **4.49** | 103.07 |
| | Total | **18.06** | 187.17 |

## E  SYSTEM EFFICIENCY AND SCALABILITY

**Graph Operation Cost Analysis.** We analyze the graph operation cost across all datasets. Table 4 compares the graph operation overhead between GNNUpdater and DGL (v2.4.0). Both systems employ UVA-based neighbor finding and GPU feature caching (caching features for 1e4 nodes and 1e7 edges). Our system achieves efficient graph operations through three key optimizations: (1) Block-based streaming graph storage reduces graph update time by 82-88% across datasets by eliminating the need for full graph reconstruction; (2) Our GPU-based neighbor finder leverages 22.24 MB of metadata stored on the GPU, achieving comparable or better performance despite using a flexible linked list data structure rather than DGL's static, compact Compress Sparse Row (CSR) representation; and (3) Optimized cache management for feature loading cuts access time by 95-99% across datasets. Together, these enhancements yield substantial reductions in graph operation overhead compared to DGL: 92.5% for `tgbn-genre`, 80.0% for `tgbn-reddit`, and 90.4% for `tgbn-token`.

**Multi-GPU Scaling.** We evaluate GNNUpdater's scalability on two additional large-scale datasets: GDELT Zhou et al. (2022), a global event interaction network (17K nodes, 191M edges, 33 GB

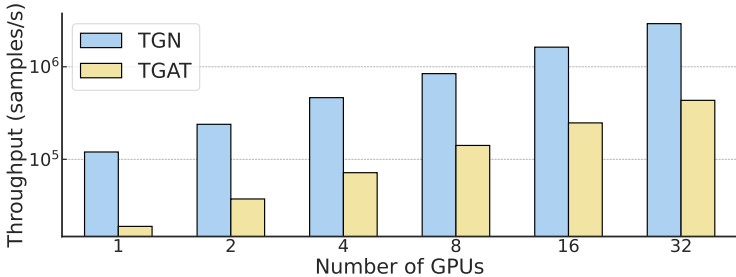

Figure 11: Training throughput scaling with different numbers of GPUs on the GDELT dataset. GNNUpdater maintains near-linear scaling within single machine (1-8 GPUs) and high scaling efficiency (>70%) across multiple machines.

Table 5: Multi-machine training performance comparison using 32 GPUs. GNNUpdater achieves faster graph construction and higher training throughput than DGL.

| Metric | DGL | GNNUpdater | Improvement |
|---|---|---|---|
| Graph Building Time (s) | 7778 | 5246 | 1.32× |
| **Throughput (samples/s)** | | | |
| GraphSAGE | 124.1k | 181.6k | 1.46× |
| GAT | 146.9k | 193.7k | 1.32× |

of edge features), and MAG Zhou et al. (2022), an academic graph (122M nodes, 1.3B edges, 175 GB of node features). All experiments are conducted on Amazon EC2 g4dn.metal instances, each equipped with 8 T4 GPUs. Using GDELT—which fits in single-machine memory—we evaluate the training throughput of TGN Rossi et al. (2021) and TGAT Xu et al. (2020a) with varying numbers of GPUs from 1 to 32. As shown in Figure 11, GNNUpdater achieves near-linear speedup on a single machine and maintains 71.9% (TGN) and 76.2% (TGAT) scaling efficiency of the ideal linear-scaling performance on 32 GPUs.

**Distributed Training.** For graphs exceeding single-machine capacity, we evaluate distributed training on partitioned MAG graph using 32 GPUs. As shown in Table 5, GNNUpdater achieves up to 1.46× training speedup compared to DGL while reducing METIS graph partitioning Karypis & Kumar (1998) and construction time by 1.32×. We further validate scalability on a synthetic LDBC graph (5.1B edges) Angles et al. (2020) distributed across 8 machines with TGN - GNNUpdater maintains efficient training with 556k samples/second throughput. These results demonstrate our system optimizations enable effective scaling from single GPU to large distributed deployments.

# F   LLM Usage Declaration

During the preparation of this manuscript, we utilized a large language model (LLM) as a writing assistant, primarily to improve the paper's clarity and positioning. The LLM's role involved assisting with reframing the narrative in the introduction and related work, clarifying methodological descriptions, and extensive copy editing to improve readability. The core research ideas, experimental design, implementation, and results were conceived and generated by the human authors; the LLM served as an assistive tool for articulating and refining the presentation of this pre-existing work.

