# OpenReview forum: "GNNUpdater: Adaptive Self-Triggered Training Framework on Dynamic Graphs"
_ICLR.cc/2026/Conference — Submitted to ICLR 2026_

### Official Review · Reviewer_QMNz · 2025-10-28

**Soundness:** 2
**Presentation:** 2
**Contribution:** 3
**Rating:** 6
**Confidence:** 4

**Summary:**

This paper studies an interesting question: when should the graph model be updated as the graph evolves? The proposed method first predicts the performance of the target nodes by measuring the embedding shifts and mapping the shift score and other graph features to a performance score. Based on the expected performance score, a threshold is set to determine whether the node is problematic. These problematic labels are then diffused by label propagation. It is finally determined whether to update the graph model based on the final ratio of "problematic" nodes.

**Strengths:**

S1. The topic is interesting and applicable to real-world applications.
S2. The update method is simple, intuitive, and effective.

**Weaknesses:**

- W1. The proposed method is not end-to-end and relies on numerous predefined hyperparameters—such as $\epsilon$, the $0.5$ term in $r_t$, and $\phi$. These choices are pivotal to the model’s performance, yet selecting them can be costly. Moreover, it is unclear whether fixed values will remain valid as the graph evolves, since the underlying dynamics may change.

- W2. The method should be compared with approaches that explicitly model graph evolution, such as JODIE, TGN, and DyGFormer. These methods naturally incorporate recent neighbors and typically do not require retraining for updates; although JODIE and TGN maintain memories, updating memory is efficient.

- W3. Another relevant line of work focuses on discrete-time dynamic graphs and employs update mechanisms similar to those proposed here (e.g., InstantGNN [R1]). Such methods should be included as baselines.

[R1] Instant Graph Neural Networks for Dynamic Graphs. KDD'22.

**Questions:**

Q1. In Eq (3), how do the four features {log(num_nodes), log(num_edges), t, deg(v)} affect the performance of the mapping function?

---

> ### Author Response · Authors · 2025-11-25
> **Rebuttal by Authors**
>
> Dear Reviewer,
>
> We sincerely thank you for the positive assessment and are greatly encouraged by your recognition of our work as interesting and applicable to real-world applications, as well as your appreciation of our method as simple, intuitive, and effective; below, we provide detailed responses to your questions.
>
> **W1: “The proposed method is not end-to-end and relies on numerous predefined hyperparameters.”**
>
> Thank you for the insightful comment. The use of explicit thresholds is a deliberate system design choice rather than a limitation arising from the method not being end-to-end.
>
> 1. **Interpretability & Operational Control.** As discussed in §3.2, thresholds such as ε represent application-level requirements (e.g., an SLA that mandates NDCG@10 ≥ 0.45). They provide practitioners with transparent control over the cost–performance trade-off—capabilities that fully end-to-end policies lack. The 0.5 cutoff in rt is not tuned but a standard majority-vote boundary.
> 2. **Stability & Low Sensitivity.** Our sensitivity analysis in Figure 10 shows that GNNUpdater performs consistently across a broad range of ε and φ values, demonstrating that expensive fine-grained tuning is not required.
> 3. **Adaptation to Evolving Graphs.** While ε and φ remain fixed to reflect stable quality standards, the Performance Predictor is continuously retrained online (§3.1). This ensures that the drift–performance mapping automatically adapts to changing graph dynamics, preserving the validity of the trigger logic even as the underlying distribution evolves.
>
> In summary, these thresholds should be viewed as interpretable, domain-grounded control knobs that offer practitioners operational clarity, rather than brittle hyperparameters. Coupled with the continually updated Performance Predictor, they provide a robust and adaptive mechanism specifically designed to handle the shifting dynamics of large-scale evolving graphs.
>
> **W2 & W3: “The method should be compared with approaches that explicitly model graph evolution (JODIE, TGN) or discrete-time dynamic graphs (InstantGNN).”**
>
> Thank you for highlighting this important point. We appreciate the opportunity to clarify how GNNUpdater relates to dynamic GNN methods such as InstantGNN, EvolveGCN, and ROLAND.
>
> **1. Orthogonal Axes: “How to learn” vs. “When to update”.**
>
> Methods like ROLAND, EvolveGCN, and InstantGNN focus on *architectural innovations* for dynamic GNNs—how the model incorporates evolving topology or temporal signals.
>
> In contrast, **GNNUpdater addresses a different axis: the *update policy***—deciding *when* these architectures should be retrained under streaming data with label delay.
>
> In practice, these architectures are normally trained either continuously or at fixed intervals. GNNUpdater is designed as a **model-agnostic wrapper** to replace that fixed schedule with an adaptive trigger. Therefore, they are *compatible backbones* rather than competing baselines.
>
> **2. Evidence using TGAT as a representative dynamic GNN backbone.**
>
> To demonstrate this compatibility, we already evaluate GNNUpdater on TGAT (Xu et al., 2020), which is architecturally similar in spirit to EvolveGCN/ROLAND in that it explicitly models temporal-topological dependencies.
>
> As shown in Figure 4, applying GNNUpdater to TGAT yields **up to 6.0× reduction in training cost** on tgbn-genre while maintaining or improving accuracy over periodic updates. This confirms that GNNUpdater can provide *added value even when paired with advanced dynamic GNNs.*
>
> **3. Why we did not exhaustively combine with all dynamic GNNs.**
>
> While InstantGNN, EvolveGCN, and ROLAND differ in architectural details, **their training schedules follow the same pattern**: update whenever data arrives or at fixed intervals. Because GNNUpdater optimizes this schedule in a model-agnostic manner, combining it with each individual architecture would yield qualitatively similar outcomes to TGAT + GNNUpdater.
>
> Nevertheless, we agree that this compatibility should be stated more explicitly. In the revision, we will update Section 4 to clarify:
>
> - These dynamic GNNs form *orthogonal, compatible backbones*.
> - GNNUpdater can be applied to them to optimize update timing.
> - TGAT serves as a representative demonstration.
>
> We thank the reviewer again for this helpful suggestion and will revise the final version accordingly.

---

> > ### Author Response · Authors · 2025-11-25
> >
> > **Q1: ”In Eq (3), how do the four features {log(num_nodes), log(num_edges), t, deg(v)} affect the performance of the mapping function?“**
> >
> > Thank you for this insightful question. While $\text{drift}(v)$ is the primary signal reflecting changes in the input distribution $p(x)$, the auxiliary features  $\\{\log(\text{num\\_nodes}),\log(\text{num\\_edges}),t,\text{deg}(v)\\}$ provide essential structural and temporal context that determines how drift impacts downstream performance $p(y|x)$:
> >
> > - **$\log(\text{num\\_nodes}) \log(\text{num\\_edges)}$** capture global graph growth and density shifts. As the graph expands or contracts, the same magnitude of embedding change may correspond to different performance implications.
> > - $t$ models temporal aging effects and periodic patterns in dynamic systems (e.g., weekly seasonality or gradual model staleness).
> > - **$\text{deg}(v)$** encodes node importance: high-degree hub nodes react differently to drift compared to low-degree tail nodes, influencing aggregated quality measurements.
> >
> > As noted in §3.1, the relationship between these contextual factors and performance is highly **non-linear**. Table 2 shows that Random Forest—capable of capturing these interactions—substantially outperforms a linear model (up to **27% error reduction**), confirming the importance of incorporating these auxiliary features.

---

### Official Review · Reviewer_6w7A · 2025-10-29

**Soundness:** 2
**Presentation:** 2
**Contribution:** 2
**Rating:** 4
**Confidence:** 3

**Summary:**

This work presents GNNUpdater, an adaptive framework for deciding when to update GNNs on dynamic graphs. It addresses label delay and hidden drift by introducing a performance predictor that estimates model degradation without ground-truth labels, using shifts in node embeddings as an indicator. A graph-aware trigger then propagates “problematic” node signals via graph propagation and triggers updates when degradation becomes widespread. Notably, they declare that they implement their work in a distributed streaming-GNN library, including a custom dynamic graph storage system to reduce operational overhead.

**Strengths:**

According to the authors, this is the first work to solve the problem of when to update models in the dynamic graph domain. Meanwhile, the approach features a simple yet well-motivated design tailored to graph data, and the explanations are sound. In addition, They release a distributed streaming-GNN library, which provides a tangible code contribution to the community.

**Weaknesses:**

1. The evaluation part compares against only three categories of existing update-trigger methods (and their straightforward variants) from previous work, which may not be sufficient. There are many new works of update triggers like:

[1] Wan, Ke, Yi Liang, and Susik Yoon. "Online drift detection with maximum concept discrepancy." Proceedings of the 30th ACM SIGKDD Conference on Knowledge Discovery and Data Mining. 2024.

[2] Lu, Pengqian, et al. "Early concept drift detection via prediction uncertainty." Proceedings of the AAAI Conference on Artificial Intelligence. Vol. 39. No. 18. 2025.

[3] Florence, Regol, et al. "When to retrain a machine learning model." arXiv preprint arXiv:2505.14903 (2025).

2. The work lacks theoretical or mathematical analysis to justify the proposed approach.

3. The evaluation omits several related baselines such as InstantGNN, EvolveGCN, and ROLAND. Despite authors’ claim of orthogonality, it is difficult to assess whether the proposed approach provides genuine added value to the field, and the work lacks any attempt to combine it with these existing methods.

**Questions:**

Please refer to the weaknesses.

---

> ### Author Response · Authors · 2025-11-25
> **Rebuttal by Authors**
>
> Dear Reviewer,
>
> We sincerely thank you for the thoughtful review and greatly value your recognition of this work as the first attempt to address the GNN update timing problem, as well as your appreciation of our distributed streaming-GNN library as a tangible contribution to the community; below, we provide detailed responses to your questions.
>
> **Q1: “The evaluation part compares against only three categories of existing update-trigger methods...”**
>
> Thank you for bringing these relevant and recent works ([1]–[3]) to our attention. We agree that they represent strong progress in update triggering strategies for **general i.i.d. data streams**, and we will incorporate them into our discussion.
>
> **However, our problem setting fundamentally differs from theirs**: we study update timing for **dynamic graphs**, where **structural dependencies** lead to *Hidden Drift*, i.e., local changes in one node propagate through the topology, degrading the embeddings of multi-hop neighbors even if their own features remain stable. Methods designed for i.i.d. data or sequential streams process samples independently and therefore cannot capture structural degradation.
>
> **Relation to Suggested Works and Existing Baselines**
>
> While we did not explicitly cite these very recent papers, their underlying methodological categories are already covered and compared in our evaluation. We demonstrate why these categories fall short in graph settings:
>
> | **Reviewer Suggested** | **Category** | **Our Existing Baseline Equivalent** | **Key Limitation on Graphs (Our Findings)** |
> | --- | --- | --- | --- |
> | **MCD [1]** | Distribution shift (Discrepancy) | **MMD** (Table 1, Fig. 4) | **Scalability & Sensitivity:** Computationally expensive ($O(N^2)$), causing OOM on large graphs (*tgbn-token*); shows weak correlation with performance (-0.40 vs. -0.80). |
> | **PUDD [2], UPF [3]** | Predicted performance / Uncertainty | **PerfDrop-PP** & **No-Propagation** (Fig. 5) | **Lack of Structural Awareness:** Relying on prediction without label propagation fails to detect Hidden Drift, incurring **1.84× higher training costs** to match GNNUpdater's performance. |
>
> **Key Takeaway:** Our ablation study (Fig. 5) explicitly demonstrates that **graph-aware propagation**—which diffuses "problematic" signals to neighbor nodes—is the critical component for detecting Hidden Drift. This mechanism is absent in general stream triggers like [1]–[3], explaining why structure-agnostic methods perform poorly on graphs despite being SOTA on i.i.d. streams.
>
> **Revision Plan:** We will:
>
> - Add [1]–[3] in the Related Work section under “General Update Triggering Methods”;
> - Add a concise discussion clarifying why **graph-aware triggers are required** to handle structural dependencies and Hidden Drift.

---

> > ### Author Response · Authors · 2025-11-25
> >
> > **Q2: “The work lacks theoretical or mathematical analysis...”**
> >
> > Thank you for this comment. We appreciate your interest in deeper guarantees, especially given recent advances in update timing for i.i.d. settings (e.g., [3]). At the same time, GNNUpdater targets **dynamic graphs**—a setting where distribution shift arises from **structural evolution, label delay, and multi-hop dependency propagation**. These factors fundamentally complicate the derivation of full regret or convergence bounds typically seen in i.i.d. frameworks.
> >
> > While deriving closed-form regret bounds in this complex setting remains highly challenging, **GNNUpdater is strictly grounded in principled formulations rather than heuristics.** We provide mathematical grounding for all core components:
> >
> > 1. **Decision-Theoretic Formulation**: Section 2 formally defines the update-timing problem as an optimization objective minimizing the sum of cumulative task loss and training cost (Eq. 1). GNNUpdater serves as a practical approximation to this optimal stopping problem under the constraints of label delay.
> >
> > 2. **Statistical Grounding for Surrogate Signals**: In Section 3.1 and Table 1-2, we provide rigorous quantitative evidence of a strong, stable correlation between our embedding drift metric and ground-truth performance (Pearson $r$ up to -0.96). This provides the empirical grounding for using drift as a reliable surrogate loss signal.
> >
> > 3. **Well-Founded Diffusion Mechanics**: The graph-aware trigger leverages the diffusion dynamics of classical Label Propagation (Zhu & Ghahramani, 2002). By using this theoretically well-understood mechanism, we capture the structural propagation of performance degradation ("hidden drift") in a way that local heuristics cannot.
> >
> > Overall, while deriving full theoretical guarantees (e.g., regret bounds with delayed and graph-structured feedback) is extremely challenging for large-scale non-stationary graphs, we believe the above components provide a solid theoretical foundation appropriate for a systems-oriented dynamic-graph contribution. Extending formal guarantees—potentially drawing on ideas from the i.i.d. retraining literature—to graph-coupled settings is a promising direction for future work.
> >
> > **Q3: “The evaluation omits several related baselines such as InstantGNN, EvolveGCN, and ROLAND…”**
> >
> > Thank you for highlighting this important point. We appreciate the opportunity to clarify how GNNUpdater relates to dynamic GNN methods such as InstantGNN, EvolveGCN, and ROLAND.
> >
> > **1. Orthogonal Axes: “How to learn” vs. “When to update”.**
> >
> > Methods like ROLAND, EvolveGCN, and InstantGNN focus on *architectural innovations* for dynamic GNNs—how the model incorporates evolving topology or temporal signals.
> >
> > In contrast, **GNNUpdater addresses a different axis: the *update policy***—deciding *when* these architectures should be retrained under streaming data with label delay.
> >
> > In practice, these architectures are normally trained either continuously or at fixed intervals. GNNUpdater is designed as a **model-agnostic wrapper** to replace that fixed schedule with an adaptive trigger. Therefore, they are *compatible backbones* rather than competing baselines.
> >
> > **2. Evidence using TGAT as a representative dynamic GNN backbone.**
> >
> > To demonstrate this compatibility, we already evaluate GNNUpdater on TGAT (Xu et al., 2020), which is architecturally similar in spirit to EvolveGCN/ROLAND in that it explicitly models temporal-topological dependencies.
> >
> > As shown in Figure 4, applying GNNUpdater to TGAT yields **up to 6.0× reduction in training cost** on tgbn-genre while maintaining or improving accuracy over periodic updates. This confirms that GNNUpdater can provide *added value even when paired with advanced dynamic GNNs.*
> >
> > **3. Why we did not exhaustively combine with all dynamic GNNs.**
> >
> > While InstantGNN, EvolveGCN, and ROLAND differ in architectural details, **their training schedules follow the same pattern**: update whenever data arrives or at fixed intervals. Because GNNUpdater optimizes this schedule in a model-agnostic manner, combining it with each individual architecture would yield qualitatively similar outcomes to TGAT + GNNUpdater.
> >
> > Nevertheless, we agree that this compatibility should be stated more explicitly. In the revision, we will update Section 4 to clarify:
> >
> > - These dynamic GNNs form *orthogonal, compatible backbones*.
> > - GNNUpdater can be applied to them to optimize update timing.
> > - TGAT serves as a representative demonstration.
> >
> > We thank the reviewer again for this helpful suggestion and will revise the final version accordingly.

---

### Official Review · Reviewer_1SUZ · 2025-10-30

**Soundness:** 3
**Presentation:** 2
**Contribution:** 3
**Rating:** 4
**Confidence:** 3

**Summary:**

This paper introduces GNNUpdater, an adaptive training triggering framework for dynamic graphs, designed to address the "when to update" challenge for Graph Neural Networks (GNNs) in continual learning environments. The framework employs a performance predictor based on node embedding drift to estimate model performance, along with a graph-aware update trigger to detect performance degradation.

**Strengths:**

This study addresses the "when to update" problem in dynamic graph learning by proposing a performance predictor and a graph-aware update trigger. The former estimates task performance without ground-truth labels. The latter monitors model quality across the entire graph.

This work develops a block-based streaming graph storage system and CUDA-accelerated GPU neighbor finder to support incremental updates and fast neighborhood sampling.

This study conducts a comprehensive evaluation on multiple real-world temporal graph benchmarks to validate the effectiveness of the proposed method.

**Weaknesses:**

In GNNUpdater, the calculation of global drift relies on the reference embeddings, $\mathbf{H}_{ref}$. However, these references are generated from a full-graph inference right after the last model update. If the interval between two updates is too long, this reference point itself could become outdated. What would happen to the performance in this case?

In Section 3.2, label propagation is used based on the graph structure. This implicitly assumes that performance degradation propagates through the graph like a label. However, the way a GNN's aggregation mechanism works might not match how actual performance issues spread. How can the validity of this operation be justified? For instance, a node connected to a problematic node isn't necessarily problematic itself. This seems somewhat questionable.

Appendix D.4 shows that different datasets require different parameter tuning ranges, which affects the generalizability of the parameter settings.

Some notations in the paper are not clearly defined, making them hard to understand. Examples include p(x) and p(y|x) in Section 2, and $C_{train, t}$ in Equation 1.

**Questions:**

Please refer to the weaknesses.

---

> ### Author Response · Authors · 2025-11-25
> **Rebuttal by Authors**
>
> Dear Reviewer,
>
> We sincerely thank you for the detailed review and constructive feedback, and greatly appreciate the recognition of our novel system design and comprehensive evaluation; below, we provide detailed responses to your questions.
>
> **Q1: “If the interval between two updates is too long, the reference embedding `H_ref` could become outdated. What would happen to the performance in this case?”**
>
> We appreciate this insightful observation regarding the potential staleness of the reference embeddings. While it is true that $H_{ref}$ is fixed between GNN updates, our framework is explicitly designed to handle this via two complementary mechanisms:
>
> 1. **Implicit Stability**: A long interval between updates occurs only if the calculated drift scores (Eq. 2) remain low. Mathematically, low drift scores imply that the current node embeddings $H_t$ are still close to $H_{ref}$ in the vector space. Therefore, if the system chooses not to update, it serves as evidence that $H_{ref}$ remains a valid anchor for the current data distribution.
> 2. **Predictor Recalibration**: Crucially, even if the semantic meaning of the distance to $H_{ref}$ changes over time (e.g., due to slow environmental shifts), our Performance Predictor compensates for this. As noted in Section 3.1, the predictor is periodically retrained using the most recent labels (a fast process, e.g., 0.25s). This ensures the mapping function $f(x)$ continuously adapts. If $H_{ref}$ becomes "stale"—such that even small drifts begin to correlate with larger performance drops—the retrained regressor will learn this new relationship and lower the predicted performance scores, thereby triggering the necessary GNN update.
>
> In summary, GNNUpdater does not blindly rely on $H_{ref}$; it relies on a **dynamically calibrated relationship** between embedding shifts (relative to $H_{ref}$) and task performance.
>
> **Q2: “The justification for using label propagation for performance degradation is weak...”**
>
> We appreciate your scrutiny on this heuristic. The validity of using Label Propagation (LP) is mathematically justified by its structural equivalence to the GNN's own aggregation mechanism.
>
> Standard GNNs (like GCN) update node embeddings by aggregating neighbors via a normalized adjacency matrix $S_t$. Our LP mechanism uses the **exact same** **operator** to diffuse degradation risk.
>
> - **GNN Aggregation**: $H^{(l+1)} = \sigma(S_t H^{(l)} W)$
> - **Our LP Update**: $y^{(k+1)} = \alpha S_t y^{(k)} + (1-\alpha)y^{(0)}$
>
> Since both processes rely on $S_t$, LP acts as a **linear approximation** of the GNN's receptive field. If a neighbor $u$ is "problematic" (drifting heavily), its degraded embedding propagates to target $v$ via $S_t$ in the GNN; similarly, its high risk score propagates to $v$ via $S_t$ in our LP module.
>
> Moreover, we agree that a connected node is not necessarily problematic. This is why our LP produces continuous scores $y_v \in [0,1]$ rather than binary labels. A score of $y_v=0.3$, for example, signifies a "risk probability" derived from the neighborhood, not a definitive failure. As defined in Section 3.2, we aggregate these soft scores into a global metric ($r_t$) to make a robust decision based on systemic risk rather than individual node noise.
>
> **Empirical Validation**: Our ablation study (Figure 5) confirms that removing this structure-aware propagation significantly harms performance (e.g., large drops on *tgbn-token*), validating that capturing this dependency is essential for detecting hidden drift.

---

> > ### Author Response · Authors · 2025-11-25
> >
> > **Q3: “Different datasets require different parameter tuning ranges, which affects the generalizability of the parameter settings.”**
> >
> > Thank you for the thoughtful comment. We acknowledge that the tuning ranges vary across datasets; however, we believe this reflects the diverse temporal and structural characteristics of the TGB benchmarks rather than a limitation of our method’s generalizability.
> >
> > **First**, as discussed in Section 3.2, the threshold $\phi$ acts as an **operational lever (knob)** to navigate the performance–cost trade-off (Eq. 1). Different domains (e.g., music recommendation vs. crypto transactions) naturally necessitate different operating points due to varying drift speeds and cost sensitivities.
> >
> > **Second**, our sensitivity analysis in Figure 10 demonstrates that GNNUpdater remains robust across broad intervals within each setting, meaning that locating a functional operating region is straightforward in practice.
> >
> > **Finally**, we emphasize that this need for calibration is universal, not specific to our method. As detailed in Appendix D.4, baseline methods exhibit even larger variances in their tuning ranges. For instance, the PerfDrop baseline requires searching small thresholds (0.5%–15%) for *tgbn-genre* but much larger thresholds (10%–50%) for *tgbn-token*. This confirms that dataset-specific calibration is inherent to these datasets.
> >
> > Overall, the generality of our framework lies in its adaptive decision-making mechanism rather than in relying on one universal hyperparameter. Our empirical results further show that this mechanism transfers effectively across diverse dynamic graph domains.
> >
> > **Q4: “Some notations in the paper are not clearly defined...”**
> >
> > We thank you for pointing this out and apologize for the lack of clarity. We will revise Section 2 to ensure all notations are explicitly and prominently defined.
> >
> > - **$p(x)$ and $p(y|x)$:** We will explicitly clarify that within our context, $x$ represents the input graph data (topology and features) and $y$ represents the labels. Thus, $p(x)$ refers to the input distribution (related to structural/feature drift), and $p(y|x)$ refers to the conditional distribution of labels (related to semantic drift).
> > - **$C_{train, t}$:** We apologize if this definition was easily overlooked. It is currently introduced in Section 2 (paragraph *“GNN Inference and Training...”*) as the computational cost of the fine-tuning operation at time $t$. We will move this definition to a clearer position or add a notation table to improve readability.

---

### Official Review · Reviewer_EV8x · 2025-10-31

**Soundness:** 3
**Presentation:** 4
**Contribution:** 3
**Rating:** 6
**Confidence:** 3

**Summary:**

This paper aims to address the problem of deciding when to update a GNN to balance model freshness and computational cost. Existing methods struggle with label delay (ground truth arrives late) and hidden drift (structural changes propagate across multi-hop neighbors). GNNUpdater addressed the problem by introducing two core components: (1) a performance predictor that estimates model quality without labels by tracking embedding shifts between current and reference node representations using a neighbor-aware drift metric; and (2) a graph-aware update trigger that applies label propagation to detect widespread degradation before major performance drops.

**Strengths:**

1. The paper tackles a practically critical question of when to update GNNs. The motivation is solid and strongly grounded in real-world industrial scenarios, where retraining is expensive and labels arrive with delay.
2. The paper is well-structured and easy to follow. Problem formulation, methods, and experiments are clear, figures effectively support the narrative.
3. Empirical results justified the effectiveness of proposed approach.

**Weaknesses:**

All experiments are on the node affinity prediction. It would be better to also explore whether the method works for other common tasks on graphs.

**Questions:**

1. On “label delay” — is this phenomenon specific to graph data, or also common in other data modalities? If it also exists in non-graph domains, how is it typically handled there and can the solutions be adapted directlt to graphs? If it is more severe in graphs, what graph-specific properties cause label delay (e.g., multi-hop dependency, verification latency, fraud propagation)?

2. In your drift formulation, node drift is computed using v and its neighbors N(v). But on a dynamic graph, nodes and edges appear/disappear, so N(v) changes over time. Is N(v) taken from the current graph structure, or the reference graph at last update? How do you handle cases where a previous neighbor disappears?

---

> ### Author Response · Authors · 2025-11-25
> **Rebuttal by Authors**
>
> Dear Reviewer,
>
> We sincerely thank you for the recognition and constructive feedback, and truly appreciate your positive comments on our solid motivation, well-structured presentation, and effective empirical results; below, we provide detailed responses to your questions.
>
> **Q1 (1): “On ‘label delay’... is this phenomenon specific to graph data... how is it typically handled and can the solutions be adapted directly to graphs?”**
>
> Thanks for this insightful question. Label delay is a ubiquitous challenge in machine learning, not limited to graphs. It frequently appears in online advertising (conversion delay) and financial fraud detection (investigation latency). In traditional non-graph domains (i.e., I.I.D. data), this is typically handled by:
>
> - **Monitoring Distribution Shifts**: Detecting degradation proxies by monitoring shifts in input features ($P(X)$) or output predictions ($P(\hat{Y})$).
> - **Heuristic Proxies**: Using faster signals (e.g., short-term interaction rates) to approximate long-term goals, though these are often domain-specific and noisy.
>
> These methods cannot be adapted effectively to dynamic graphs due to the non-I.I.D. nature of the data:
>
> - **In Non-Graph ML**: Data points are independent. A distribution shift in user $A$ acts locally and does not immediately alter the representation of user $B$.
> - **In Graph ML**: Due to the message-passing mechanism of GNNs, a change in one node (e.g., a new transaction) propagates through the structure, altering the embeddings of its 1-hop and 2-hop neighbors.
>
> **Q1 (2): "If it is more severe in graphs, what graph-specific properties cause label delay (e.g., multi-hop dependency, verification latency, fraud propagation)?"**
>
> The source of the delay is typically application-level (e.g., "verification latency" for manual fraud reviews), not graph-specific. However, graph properties drastically worsen the consequence of this delay. While the system is waiting for a delayed label (due to "verification latency"), "fraud propagation" is already silently degrading embeddings across the graph via "multi-hop dependency," causing widespread, undetected performance drops.
>
> **Q2: “In your drift formulation... Is N(v) taken from the current graph structure, or the reference graph at last update? How do you handle cases where a previous neighbor disappears?”**
>
> Thank you for the clarifying question. The neighborhood $N(v)$ is always taken from the **current graph structure at time $t$** to assess the model on the most recent data. Disappearing neighbors are simply excluded from the current calculation, while new neighbors are included. This design is critical for two reasons:
>
> 1. **Consistency with Inference:** The GNN's forward pass aggregates information solely from the current topology. Using an outdated neighborhood would define drift based on nodes that structurally no longer contribute to the node's embedding ($h_v^t$), creating a misalignment between the monitoring metric and the model's actual receptive field.
> 2. **Avoidance of Artificial Drift:** Including disappeared neighbors would introduce "ghost" drift signals from nodes that are architecturally irrelevant to the current prediction.
>
> By strictly aligning the drift calculation with the GNN's aggregation scope, our approach effectively captures the input shifts that matter to the model, validated by the strong drift-performance correlation shown in Table 1.
>
> **W1: “All experiments are on the node affinity prediction...”**
>
> We thank the reviewer for this insightful suggestion. While our current evaluation focuses on node affinity prediction, the GNNUpdater framework is modular and designed to be **task-agnostic**.
>
> Currently, there is a lack of large-scale temporal graph datasets with sufficient node-level labels in the literature (Huang et al., 2023); hence, our evaluation is restricted to the node affinity prediction task.
>
> However, GNNUpdater is inherently task-agnostic. For instance, to support dynamic link prediction, we only need to retrain the Performance Predictor (§3.1) to map embedding drift to a link-based metric (e.g., MRR) instead of a node-ranking metric (NDCG). The core drift detection and trigger mechanisms (§3.2) remain unchanged.
>
> Future work includes extending GNNUpdater to other temporal graph learning tasks—such as dynamic node classification and dynamic link prediction.

---

### Meta-Review · Area_Chair_CcUd · 2026-01-07

**Summary:**

This work proposed GNNUpdater, a framework that decides how to re-train GNNs adaptively on dynamic graphs, by using embedding drift metrics to predict performance degradation, and propagating degradation to help inform an update trigger.  Reviewers found the problem setting interesting and practical (EV8x, QMNz), appreciated simple design (6w7A, QMNz), and noted the strong results and scalable library (EV8x, 1SUZ). However, the submission has several important weaknesses that reviewers also pointed out:

- the evaluation scope is narrow, focusing on node affinity prediction and excluding other dynamic graph tasks (EV8x)

- multiple reviewers pointed out that several baselines and alternate trigger-based/update-based works should have been discussed or positioned against (6w7A, QMNz) but authors did not sufficiently convince reviewers during rebuttal that their positioning is indeed totally fair. several of the techniques proposed in this work don't seem to be too different from non-graph based drift detection methods, and despite the setting being different, the core methodology does not seem to be (except for the update trigger, which itself has questionable comparison with other triggers) (AC)

- the approach relies on multiple manually chosen thresholds and heuristics, which raises concerns about sensitivity to dynamic graph data (1SUZ, QMNz)

- the work lacks formal theoretical analysis, with only partial empirical justification for the drift mechanism (6w7A)

Overall, despite the positives, this paper fell short of an accept recommendation at this time; I encourage authors to iterate for a future submission.

**Reviewer Concerns:**

See above.

**Reviewer Scores:**

EV8x: 6
1SUZ: 4->4/5
6w7A: 4->4
QMNz: 6

---

### Decision · Program_Chairs · 2026-01-26

Reject